# Prevalence and correlates of metabolic syndrome in patients with initial-treatment and drug-naïve bipolar disorder: A large sample cross-sectional study

Yilin Fang[1☯], Bingchuan Yan[1☯], Zhihua Liu[2]*, Lin Zhang [3]*

1 The First Affiliated Hospital of Luohe Medical College (Luohe Central Hospital), Luohe, Henan, China, 2 Department of Psychiatry, The Fourth People's Hospital of Nanyang, Nanyang, Henan, China, 3 Department of Psychiatry, Wuhan Mental Health Center, Wuhan, Hubei, China

☯ These authors contributed equally to this work.
* zhihua0188@wo.cn (ZL); linzhang16@qq.com (LZ)

## Abstract

Patients with bipolar disorder (BD) are frequently prone to metabolic syndrome (MetS), and their co-morbidity adversely affects patient care outcomes. This study aimed to determine the prevalence of MetS and its clinical correlates among initial-treatment and drug-naïve (ITDN) BD patients.We recruited a cohort of 841 ITDN BD patients. Socio-demographic and clinical data were collected, and patients underwent routine serological testing, which included fasting blood glucose, lipid profiles, thyroid function, and prolactin levels. Psychometric evaluations were also conducted to measure manic, depressive, and psychotic symptoms, as well as illness severity. Additionally, we utilized a transformation approach for continuous variable analysis to compute a MetS score.We found a MetS prevalence of 17.84% among the study participants. Binary logistic regression identified age, body mass index (BMI), low-density lipoprotein cholesterol (LDL-C), free tetraiodothyronine (FT4), and psychotic symptoms as significant predictors of MetS development. Further, multiple linear regression analysis indicated that advanced age was a significant predictor of higher MetS scores.The findings highlight the prevalence of MetS in ITDN BD patients and suggest that certain demographic and clinical factors are influential in the development and severity of MetS. These insights may guide the development of targeted preventive and therapeutic strategies for MetS in this patient population.

## 1. Introduction

Metabolic syndrome (MetS) is a chronic, non-infectious clinical syndrome characterized by a cluster of vascular risk factors, including insulin resistance, hypertension, abdominal obesity, impaired glucose metabolism, and dyslipidemia [1]. With the global adoption of Western dietary habits [2,3], MetS has surpassed traditional

**Data availability statement:** All relevant data are within the paper.

**Funding:** The author(s) received no specific funding for this work.

**Competing interests:** The authors have declared that no competing interests exist.

infectious diseases as a significant health threat, increasingly affecting children and adolescents [4,5]. MetS is recognized as a known risk factor that poses a significant threat to somatic health and life expectancy [6,7]. However, the detrimental effects of MetS on patients with BD go far beyond this. The co-occurrence of BD and MetS is associated with worsened executive functioning [8], adverse clinical outcomes [9], and severe cognitive impairments and brain imaging abnormalities [10]. Understanding the clinical features of MetS in BD patients during the initial-treatment and drug-naïve (ITDN) phases and identifying factors that influence its development are crucial for the effective primary prevention of MetS and improving the prognosis in BD.

Emerging evidence suggests a complex relationship between MetS and mood disorders, influenced by clinical, neurobiological, genetic, and environmental factors [11,12]. This link is especially pronounced in individuals with severe mental illnesses such as bipolar disorder (BD), making them vulnerable to MetS [13,14]. While the metabolic adverse effects of antipsychotic medications in BD patients are well-documented [15,16], research focusing on drug-naïve BD populations remains scarce. To date, only a limited number of small-scale studies have explored MetS prevalence in untreated BD patients, reporting heterogeneous rates ranging from 6.5% to 33.3% [17,18]. For instance, a study by Hung Chi et al. involving 60 drug-naïve BD patients in Taiwan reported a MetS prevalence of 16.7% [17], particularly in patients with severe psychiatric disorders who are susceptible to it [18]. These studies, though insightful, are constrained by methodological limitations such as modest sample sizes, regional specificity, and reliance on dichotomous MetS classifications, which fail to capture its severity.

This gap in the literature underscores the need for large-scale, methodologically rigorous investigations to establish robust prevalence estimates and identify clinical correlates of MetS in drug-naïve BD populations. By excluding the confounding effects of psychotropic medications, such studies could elucidate intrinsic metabolic vulnerabilities in BD, offering critical insights for early intervention. Our study directly addresses this gap by examining a large cohort of 841 ITDN BD patients, employing both categorical and continuous MetS assessments. This approach not only refines prevalence estimates but also quantifies metabolic dysfunction severity, advancing understanding of MetS pathophysiology in untreated BD.

## 2. Participants and methods

### 2.1. Subjects

The sample size was determined based on prior prevalence studies of MetS in BD cohorts. Assuming an estimated MetS prevalence of 15–25% in untreated BD patients [17,19], a minimum sample of 753 participants was required to achieve 80% power ($\alpha = 0.05$, two-tailed) for detecting prevalence differences ±5% via $\chi^2$ tests. Accounting for potential data incompleteness (~10% attrition), the target was set to 811 participants, which also ensured robust multivariate regression analyses (≥10 events per predictor variable [20]).

A total of 841 BD patients with ITDN were prospectively recruited from the Nanyang No. 4 People's Hospital and Wuhan Mental Health Centre between

10/12/2016 and 10/06/2024. All participants were consecutively enrolled upon their first hospitalization, with eligibility assessed through structured interviews and medical evaluations. Details of the data sources and the study flow are illustrated in Fig 1.

The inclusion criteria for participants were as follows:

1. Diagnosis according to the International Classification of Diseases, 10th edition (ICD-10) for any form of BD, including, but not limited to, bipolar I disorder, bipolar II disorder, manic episodes, depressive episodes, and mixed episodes.

2. First-time hospitalization with no prior exposure to antipsychotics, mood stabilizers, or antidepressants, although use of benzodiazepines was permitted.

3. Age between 18 and 60 years of either sex, and of Han Chinese ethnicity.

4. A score of ≥7 on the 17-item Hamilton Depression Scale (HAMD-17) and/or a score of ≥6 on the Young Mania Rating Scale (YMRS).

Patients were excluded if they:

1. Were under 18 or over 60 years of age.

2. Had any other psychiatric disorder such as major depressive disorder, schizophrenia spectrum disorder, schizoaffective disorder, personality disorder, substance abuse and dependence, intellectual developmental disorder, etc.

3. Had severe physical illnesses, autoimmune diseases, or acute inflammatory diseases.

4. Patients who were already using hypoglycemic, hypoglycemic, and hypolipidemic medications prior to evaluation were excluded.

5. Were pregnant or lactating.

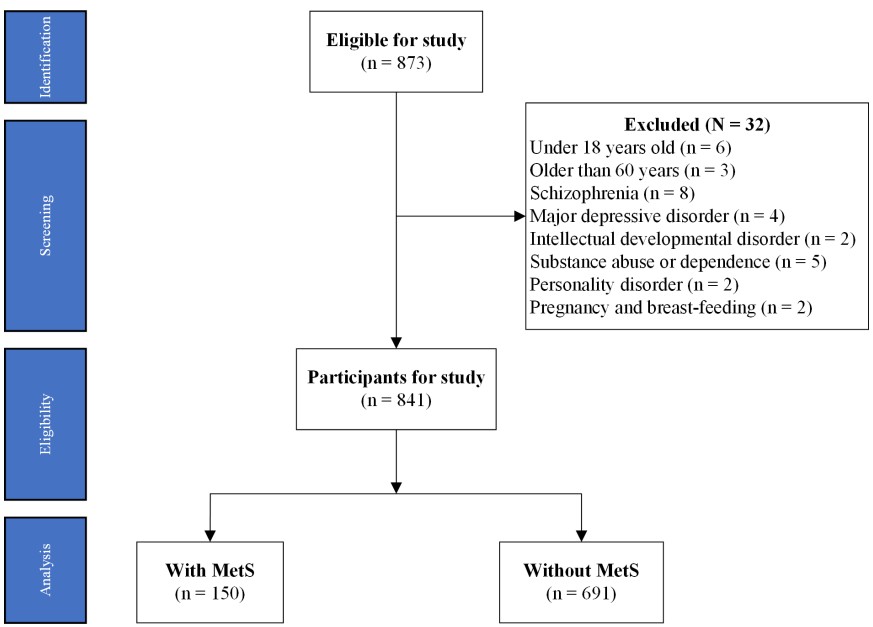

**Fig 1. Flow diagram of participants in study.** Note: MetS: metabolic syndrome.

The study protocol received approval from the Ethics Committee of the Fourth People's Hospital of Nanyang City (Nanyang Mental Health Centre) under the code 2016-BHD003. Written informed consent was obtained from all participants and/or their guardians.

## 2.2. Research design

This two-center, cross-sectional study employed a prospective design to evaluate MetS prevalence and correlates in ITDN BD patients. Baseline data were systematically collected during hospitalization, including:

**Socio-demographic and general clinical data collection.** Custom Excel forms were used to extract the following information from the electronic case system for included patients: age, age of onset, duration of disease, gender, education, marital status, height, weight, and ethnicity.

**Clinical Biochemical parameters and serological tests.** Patients were instructed to fast from 8 PM the previous night, and venous blood samples along with measurements for blood pressure and waist circumference were collected between 6 AM and 8 AM the following morning. All blood samples were immediately processed at the biochemistry laboratory of the attending healthcare facility and analyzed by 11 AM. Key indicators included renal function (uric acid, blood urea nitrogen, blood creatinine), blood lipids (total cholesterol, triglycerides, low-density lipoprotein cholesterol, high-density lipoprotein cholesterol), fasting blood glucose, thyroid function (hyroid-stimulating hormone, free triiodothyronine, free tetraiodothyronine, thyroxine, triiodothyronine) and prolactin levels.

**Clinical psychological measures.** Upon admission, patients were evaluated for depressive and manic symptoms, as well as overall illness severity using the Hamilton Depression Scale (HAMD-17) [21], the Young Mania Rating Scale (YMRS) [22], and the Clinical Global Impression Scale-Severity of Illness (CGI-SI) [23], respectively. Additionally, the positive symptom subscale (PSS) of the Positive and Negative Symptom Scale (items P1-P7) was used to assess any comorbid psychotic symptoms in patients [24]. Two attending psychiatrists, who had received uniform training and whose assessments consistently showed correlation coefficients above 0.8 between scales, conducted these evaluations.

**Diagnostic criteria for MetS.** According to the standards tailored for the Chinese population by the Chinese Diabetes Society [25], MetS is diagnosed when three or more of the following criteria are met: (1) central or abdominal obesity: waist circumference ≥90 cm for men and ≥85 cm for women; (2) hyperglycemia: fasting blood glucose ≥6.10 mmol/L or 2-hour postprandial glucose ≥7.80 mmol/L, or those diagnosed and treated for diabetes mellitus; (3) hypertension: blood pressure ≥ 130/85 mmHg or those treated for hypertension; (4) elevated fasting triglycerides (TG) ≥ 1.7 mmol/L; (5) reduced fasting high-density lipoprotein cholesterol (HDL-C) < 1.04 mmol/L.

**Quantifying MetS severity.** To evaluate MetS severity, it was necessary to convert it from a dichotomous to a continuous variable. First, the mean arterial pressure (MAP) was calculated as one-third systolic blood pressure (SBP) plus two-thirds diastolic blood pressure (DBP). Then, MetS scores were calculated for different genders among Chinese Han adults based on the recent findings of Shujuan Yang et al. [26]:

$$\text{Male: MetS score} = -2.9092 + 0.0262 \times WC + 0.3098 \times TG - 0.944 \times HDL - C + 0.0097 \times MAP + 0.0745 \times FBG$$

$$\text{Female : MetS score} = -2.4981 + 0.0199 \times WC + 0.5218 \times TG - 0.8616 \times HDL - C + 0.0110 \times MA + 0.1074 \times FBG$$

## 2.3. Data analysis

We report categorical variables using counts and continuous variables using means and standard deviations. Initially, pie charts were plotted to display the distribution of different MetS subtypes. Subsequently, independent samples t-tests and chi-square tests were used to compare clinical variables between the two subgroups (with and without MetS). A binary logistic regression model was then constructed to identify predictors of MetS development, using MetS as the outcome

variable and variables showing significant differences in univariate analyses as independent variables. Finally, a multivariate linear regression model was developed to identify factors predicting the severity of MetS, with MetS scores as the outcome variable and the independent variables identified in the binary logistic regression analyses. Data were analyzed using SPSS version 27.0, with statistical significance set at $p < 0.05$ (two-tailed).

## 3. Results

### 3.1. Prevalence and composition of MetS in the target population

Of the observed target population, 150 cases, or 17.84% (150/841), met the diagnostic criteria for MetS. Fig 2 illustrates the composition of MetS, highlighting that 76.00% of cases exhibited three component anomalies, representing the most common subtype.

### 3.2. Differences in clinical parameters between subgroups with and without MetS

Table 1 presents a comparison between the subgroups with and without co-morbid MetS, focusing on demographic data and general clinical information. The subgroup with MetS displayed significantly higher levels of several clinical indicators, such as age ($t = -2.03$, $p = 0.042$), body mass index (BMI) ($t = -4.65$, $p < .001$), low-density lipoprotein cholesterol (LDL-C) ($t = -4.31$, $p < .001$), total tetraiodothyronine ($TT_4$) ($t = -2.83$, $p = 0.005$), free tetraiodothyronine (FT4) ($t = -2.66$, $p = 0.008$), Clinical Global Impression Scale-Severity of Illness (CGI-SI) ($t = -2.56$, $p = 0.011$), and PSS ($t = -4.24$, $p < .001$). Conversely, creatinine (CRE) levels ($t = 2.60$, $p = 0.009$) and thyroid-stimulating hormone (TSH) levels ($t = 3.43$, $p = 0.001$) were lower in the MetS subgroup.

### 3.3. Factors influencing the development of MetS

In Table 2, binary logistic regression models (Backward: Wald) were constructed using MetS as the outcome variable and variables identified in previous univariate analyses (excluding MetS components and scores) as predictors. The analysis revealed that age ($B = 0.03$, $p = 0.006$, OR = 1.03, 95% CI: 1.01–1.05), BMI ($B = 0.13$, $p < .001$, OR = 1.14, 95% CI: 1.08–1.21), LDL-C ($B = 0.45$, $p = 0.009$, OR = 1.57, 95% CI: 1.12–2.20), FT4 ($B = 0.97$, $p = 0.011$, OR = 2.65, 95% CI: 1.25–5.63), and PSS ($B = 0.37$, $p < .001$, OR = 1.45, 95% CI: 1.37–1.54) significantly positively predicted MetS development. Conversely, CRE ($B = -0.04$, $p < .001$, OR = 0.97, 95% CI: 1.01–1.05) and $TT_4$ ($B = -0.16$, $p = 0.010$, OR = 0.85, 95% CI: 0.75–0.96) were significant negative predictors.

### 3.4. Factors influencing the severity of MetS

As depicted in Table 3, multivariate linear regression models were used to identify predictors of MetS severity, with the MetS score as the outcome variable. The results showed that age ($B = 0.01$, $t = 2.60$, $p = 0.010$, 95% CI = 0.00–0.02) and

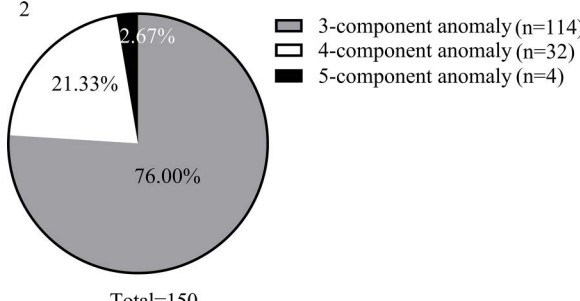

**Fig 2. Composition ratio for MetS.**

**Table 1. Between-group differences in sociodemographic and general clinical information, grouped by MetS.**

| Index | Total patients (n = 841) | MetS (n = 150) | Non-MetS (n = 691) | $t/\chi^2$ | p-value | 95%CI |
|---|---|---|---|---|---|---|
| Age – years | 27.59 ± 11.75 | 29.36 ± 12.12 | 27.21 ± 11.64 | −2.03 | 0.042* | 0.01–0.35 |
| Illness duration – months | 7.35 ± 7.11 | 8.03 ± 6.58 | 7.20 ± 7.22 | −1.30 | 0.194 | −0.05–0.29 |
| Onset age – years | 21.06 ± 8.82 | 21.69 ± 9.21 | 20.92 ± 8.73 | −0.97 | 0.333 | −0.08–0.26 |
| Gender – (n, %) | | | | 0.74 | 0.390 | 0.00–0.06 |
| Female | 440, 52.32% | 84, 56.00% | 365, 51.52% | | | |
| Male | 401, 47.68% | 66, 44.00% | 335, 48.48% | | | |
| Marital status – (n, %) | | | | 0.04 | 0.843 | 0.00–0.04 |
| With spouse | 230, 27.35% | 42, 28.00% | 188, 27.21% | | | |
| Without spouse | 611, 72.65% | 108, 72.00% | 503, 72.79% | | | |
| Educational background – (n, %) | | | | 1.18 | 0.277 | 0.00–0.07 |
| High school or below | 471, 56.00% | 90, 60.00% | 381, 55.14% | | | |
| Upper high school | 370, 44.00% | 60, 40.00% | 310, 44.86% | | | |
| MetS dimensions | | | | | | |
| WC – cm | 81.35 ± 8.81 | 85.88 ± 9.71 | 80.37 ± 8.29 | −7.14 | <.001* | −7.02--3.99 |
| FBG – mmol/L | 5.20 ± 2.42 | 5.69 ± 3.16 | 5.09 ± 2.21 | −2.22 | 0.028* | −7.19--3.82 |
| TG – mmol/L | 4.27 ± 1.57 | 1.67 ± 0.85 | 1.25 ± 0.84 | −5.56 | <.001* | −1.03--0.18 |
| HDL-C – mmol/L | 1.23 ± 0.30 | 1.10 ± 0.28 | 1.26 ± 0.30 | 6.16 | <.001* | −1.14--0.07 |
| SBP - mmHg | 124.52 ± 8.43 | 131.64 ± 8.80 | 122.98 ± 7.5 | −12.41 | <.001* | −0.57--0.28 |
| DBP - mmHg | 78.22 ± 6.16 | 83.70 ± 6.57 | 77.04 ± 5.38 | 11.61 | <.001* | −0.58--0.27 |
| MetS scores | 0.06 ± 0.63 | 0.56 ± 0.05 | −0.05 ± 0.58 | −11.45 | <.001* | 0.28–0.62 |
| BMI – kg/m² | 23.33 ± 4.35 | 24.99 ± 4.96 | 22.97 ± 4.11 | −4.65 | <.001* | −0.15–0.19 |
| BUN – mmol/L | 3.98 ± 1.27 | 3.95 ± 1.20 | 3.98 ± 1.29 | 0.32 | 0.751 | 0.10–0.44 |
| CRE – mmol/L | 63.73 ± 22.28 | 59.42 ± 13.19 | 64.65 ± 23.69 | 2.60 | 0.009* | −0.12–0.22 |
| UA – mmol/L | 375.92 ± 109.19 | 380.51 ± 104.63 | 374.94 ± 110.19 | −0.56 | 0.573 | −0.03–0.31 |
| TC – mmol/L | 4.27 ± 1.57 | 4.43 ± 1.08 | 4.23 ± 1.65 | −1.45 | 0.148 | 0.24–0.58 |
| LDL-C – mmol/L | 2.20 ± 0.72 | 2.46 ± 0.83 | 2.15 ± 0.69 | −4.31 | <.001* | 0.05–0.39) |
| TSH - uIU/mL | 2.60 ± 3.47 | 2.05 ± 1.63 | 2.72 ± 3.75 | 3.43 | 0.001* | 0.09–0.43 |
| $TT_4$ – ng/mL | 7.88 ± 2.22 | 8.35 ± 2.20 | 7.78 ± 2.22 | −2.83 | 0.005* | −0.13–0.21 |
| $TT_3$ – ng/mL | 0.98 ± 0.24 | 0.99 ± 0.25 | 0.98 ± 0.24 | −0.16 | 0.872 | −0.09–0.25 |
| $FT_3$ – pmol/L | 3.07 ± 1.23 | 3.00 ± 0.63 | 3.09 ± 1.33 | 0.84 | 0.401 | 0.07–0.41 |
| $FT_4$ – pmol/L | 1.05 ± 0.37 | 1.12 ± 0.45 | 1.03 ± 0.34 | −2.66 | 0.008* | −0.15–0.19 |
| PRL – mmol/L | 22.09 ± 28.78 | 21.55 ± 28.27 | 22.21 ± 28.91 | 0.25 | 0.800 | 0.01–0.35 |
| HAMD | 18.47 ± 2.75 | 18.90 ± 3.28 | 18.38 ± 2.61 | −1.82 | 0.071 | −0.04–0.30 |
| YMRS | 17.93 ± 1.55 | 17.75 ± 1.75 | 17.97 ± 1.5 | 1.40 | 0.162 | 0.08–0.42 |
| CGI-SI | 5.49 ± 0.60 | 5.63 ± 0.73 | 5.46 ± 0.57 | −2.56 | 0.011* | 1.30–1.74 |
| PSS | 9.38 ± 3.97 | 14.55 ± 5.29 | 8.25 ± 2.47 | −4.24 | <.001* | 0.46–0.82 |

MetS: metabolic syndrome; WC: waist circumference; FBG: fasting blood glucose; TG: triglycerides; HDL-C: high density lipoprotein cholesterol; SBP: systolic blood pressure; DBP: diastolic blood pressure; BMI: body mass index; BUN: blood urea nitrogen; CRE: blood creatinine; UA: blood uric acid; TC: total cholesterol; LDL-C: low density lipoprotein cholesterol; TSH: thyroid stimulating hormone; $TT_3$: total triiodothyronine; $TT_4$: total thyroxine; $FT_3$: free triiodothyronine; $FT_4$: free tetraiodothyronine; PRL: prolactin; HAMD: Hamilton Depression Scale; YMRS: Young Mania Rating Scale; CGI-SI: Clinical Global Impression Scale – Severity of Illness; PSS: Positive and Negative Syndrome Scale – Positive Symptom Subscale. *$p < 0.05$.

**Table 2. Identification of factors affecting the development of MetS: based on a binary logistic regression model.**

| | Coefficients | Std. error | Wald | p-value | 95% CI for EXP (B) | | |
| | B | | | | Exp(B) | Lower | Upper |
|---|---|---|---|---|---|---|---|
| Age – years | 0.03 | 0.01 | 7.60 | 0.006* | 1.03 | 1.01 | 1.05 |
| BMI – kg/m$^2$ | 0.13 | 0.03 | 20.15 | <.001* | 1.14 | 1.08 | 1.21 |
| CRE – mmol/L | −0.04 | 0.01 | 14.70 | <.001* | 0.97 | 0.95 | 0.98 |
| LDL-C – mmol/L | 0.45 | 0.17 | 6.87 | 0.009* | 1.57 | 1.12 | 2.20 |
| TSH - uIU/mL | −0.17 | 0.09 | 3.81 | 0.051 | 0.85 | 0.71 | 1.00 |
| TT$_4$ – ng/mL | −0.16 | 0.06 | 6.68 | 0.010* | 0.85 | 0.75 | 0.96 |
| FT$_4$ – pmol/L | 0.97 | 0.39 | 6.40 | 0.011* | 2.65 | 1.25 | 5.63 |
| PSS | 0.37 | 0.03 | 17.56 | <.001* | 1.45 | 1.37 | 1.54 |

BMI: body mass index; CRE: blood creatinine; LDL-C: low density lipoprotein cholesterol; TSH: thyroid stimulating hormone; TT$_4$: total thyroxine; FT$_4$: free tetraiodothyronine; PSS: Positive and Negative Syndrome Scale – Positive Symptom Subscale. *$p < 0.05$.

**Table 3. Determination of factors affecting the severity of MetS: based on multiple linear regression modelling.**

| | Coefficients | Std. error | t | p-value | 95% CI | |
| | B | | | | Lower | Upper |
|---|---|---|---|---|---|---|
| Constant | 1.09 | 0.35 | 3.14 | 0.002 | 0.40 | 1.78 |
| Age – years | 0.01 | 0.00 | 2.60 | 0.010* | 0.00 | 0.02 |
| CRE – mmol/L | −0.02 | 0.00 | −4.08 | <.001* | −0.02 | −0.01 |
| LDL – C – mmol/L | 0.18 | 0.06 | 2.93 | 0.004* | 0.06 | 0.29 |
| TT$_4$ – ng/mL | −0.05 | 0.02 | −2.00 | 0.047* | −0.09 | 0.00 |

CRE: blood creatinine; LDL-C: low density lipoprotein cholesterol; TT$_4$: total thyroxine. * $p < 0.05$.

LDL-C (B = 0.18, t = 2.93, $p$ = 0.004, 95% CI = 0.06–0.29) were risk factors for increased MetS severity. In contrast, CRE (B = −0.02, t = −4.08, $p$ < .001, 95% CI = −0.02 - −0.01) and TT$_4$ (B = −0.05, t = −2.00, $p$ = 0.047, 95% CI = −0.09–0.00) served as protective factors.

## 4. Discussion

This study provides novel insights into metabolic dysfunction in BD by uniquely focusing on ITDN patients—a critical yet understudied population that eliminates confounding effects of psychotropic medications. Our cohort (N = 841), rigorously restricted to 18–60 years, minimizes age-related metabolic heterogeneity: younger participants avoid developmental variability (e.g., adolescent hormonal fluctuations), while older adults exclude geriatric comorbidities (e.g., age-driven cardiovascular decline). By employing a dual analytical framework—traditional binary MetS classification and a continuous severity score validated for Chinese Han adults—we capture both diagnostic thresholds and subclinical metabolic gradients. Importantly, age was statistically adjusted in all models and stratified sensitivity analyses confirmed consistent associations across subgroups, reinforcing the robustness of predictors like BMI and LDL-C. This methodological rigor establishes intrinsic metabolic risk profiles in untreated BD, offering a critical baseline for early intervention strategies prior to pharmacotherapy-induced metabolic alterations.

The prevalence of MetS in our ITDN BD cohort (17.84%) is lower than rates reported in treated BD populations (27.5%–53.7%) [13,14,27], aligning with recent findings from drug-naïve studies (6.5%–33.3%) [17,18]. This discrepancy may reflect the absence of antipsychotic exposure in our cohort, a key driver of metabolic dysregulation in BD [16,28]. Notably, Mohd Ahmed et al. (2024) reported a 32.1% MetS prevalence in treated BD patients, attributing elevated risk to

antipsychotic polypharmacy and longer illness duration [13]. Similarly, Aziz et al. identified antipsychotic polypharmacy as a significant predictor of MetS in BD, independent of demographic factors [29]. Our lower prevalence underscores the impact of pharmacotherapy on metabolic risk and highlights the need for baseline metabolic profiling in untreated patients.

Our identification of age, BMI, LDL-C, FT4, and psychotic symptoms as predictors of MetS aligns with emerging evidence but also extends prior findings through novel mechanistic insights. While Aziz et al. linked psychotic symptoms to elevated MetS risk via inflammatory pathways in treated BD patients [29], our drug-naïve cohort uniquely implicates LDL-C as a stronger predictor than triglycerides, suggesting lipid dysregulation may precede antipsychotic exposure. This finding contrasts with Mohd Ahmed et al., who emphasized BMI and triglycerides in medicated cohorts [13], highlighting the importance of studying untreated populations to disentangle intrinsic metabolic vulnerabilities. Regarding thyroid dysfunction, our initial statement oversimplified the FT4-MetS link. Beyond incidental associations, thyroid hormones critically regulate metabolic homeostasis through: (1) Lipid Metabolism: FT4 upregulates hepatic lipase activity, accelerating triglyceride hydrolysis and LDL oxidation [30], consistent with our observed LDL-C elevations. (2) Insulin Signaling: Hyperthyroxinemia induces mitochondrial oxidative stress and adiponectin suppression, impairing glucose tolerance even in euthyroid BD patients [31]. (3) Autoimmune-Inflammatory Crosstalk: BD patients exhibit higher anti-thyroid antibody prevalence (e.g., anti-TPO) [32], which synergizes with pro-inflammatory cytokines to promote adipose dysfunction and central obesity [33].

Obesity, or BMI, has traditionally been an independent risk factor for MetS, and BD patients are no exception [18,34]. Our study additionally determined that an increase in BMI per unit leads to a 14% increased risk of developing MetS. Furthermore, we cannot ignore the predictive role of psychotic symptoms for MetS. Unfortunately, studies exploring this predictive ability in BD populations are relatively rare. However, patients with co-morbid MetS have been similarly observed to have more severe psychotic symptoms in patients with schizophrenia spectrum disorders and major depressive disorders, and the predictive ability of the latter for the former component has been affirmed [35–37]. Genetic studies suggest that a shared genetic etiology may be a plausible explanation for the association between the two [38,39]. The association between psychotic symptoms and MetS severity, though underexplored in drug-naïve BD, mirrors findings in schizophrenia [36,37]. This supports a shared biological substrate (e.g., dopamine-glucose metabolism interplay) across severe mental illnesses [38].

The traditional way of defining MetS based on dichotomization has some loopholes as this fails to demonstrate the degree of MetS abnormality, thus limiting its applicability in the clinical setting. To ameliorate these shortcomings, we computed MetS scores and explored in depth the factors that influence the severity oh degree. The present study found that increasing age induces deterioration and exacerbation of MetS. It is well known that MetS is also labelled as one of the classic examples of "age-related metabolic disease" [40]. A combination of aging-induced mitochondrial dysfunction, genetic and environmental factors make age a natural risk factor for the increased incidence and severity of MetS [41,42]. This has been reported in other types of severe psychiatric disorders, in addition to the patients with BD [18,34,43].

However, our study has limitations. The cross-sectional design restricts our ability to infer causality. The scarcity of research on MetS in ITDN BD populations limits the depth of our discussions, and the exclusion of adolescents, who are also affected by MetS, restricts the generalizability of our results. Future research should employ rigorous longitudinal designs to better understand these associations.

## 5. Conclusions

In summary, this study not only highlights the prevalence of MetS in BD patients with ITDN, but also elucidates clinical factors that are critical to understanding its development and severity. Thyroid dysfunction, high BMI, and co-morbid psychotic symptoms predict MetS development, and higher age implies a more worsened degree of MetS. These findings provide potential targets for MetS prevention and intervention.

## Acknowledgments

We express our gratitude to all the medical staff and patients who participated in our study, as well as to those who contributed to the diagnosis and clinical evaluation of the subjects.

## Author contributions

**Conceptualization:** Yilin Fang.

**Data curation:** Bingchuan Yan.

**Formal analysis:** Zhihua Liu.

**Methodology:** Lin Zhang.

**Supervision:** Lin Zhang.

**Validation:** Lin Zhang.

**Writing – original draft:** Yilin Fang.

**Writing – review & editing:** Bingchuan Yan, Zhihua Liu.

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
