## [Decision Letter · Decision Letter 0]

Dear Dr.  Zhang,

We look forward to receiving your revised manuscript.

Kind regards,

Dirceu Henrique Paulo Mabunda, M.D.

Academic Editor

PLOS ONE

Journal Requirements:

2. We note that your Data Availability Statement is currently as follows: “All relevant data are within the manuscript and in Supporting Information files.”

Additional Editor Comments:

After address the issues raised by the reviewers you can submit the manuscript for final decision.

Reviewers' comments:

Reviewer's Responses to Questions

**Comments to the Author**

1. Is the manuscript technically sound, and do the data support the conclusions?

Reviewer #1: Yes

Reviewer #2: Yes

2. Has the statistical analysis been performed appropriately and rigorously?

Reviewer #1: Yes

Reviewer #2: Yes

3. Have the authors made all data underlying the findings in their manuscript fully available?

Reviewer #1: Yes

Reviewer #2: Yes

4. Is the manuscript presented in an intelligible fashion and written in standard English?

Reviewer #1: Yes

Reviewer #2: Yes

Reviewer #1: Many thanks for asking me to review this very interesting paper. I am recommending some revisions. If the authors can offer a point-by-point response to each of my queries and address the issues raised, the manuscript may potentially be acceptable for publication.

Detailed comments:

Introduction:

-It would be helpful if the authors could highlight if any previous studies have investigated MetS in drug naïve patients BD. This would help demonstrate the potential value of their study by identifying gaps in the literature, that their study could potentially address i.e. please include a strong rationale for conducting the current study.

-Please include a hypothesis at the end of the introduction after the aims.

Methods:

-Were patients recruited prospectively, or was data recruited retrospectively from the patient medical records?

-Please have an operational definition for “initial-treatment and drug-naïve”, especially for “initial treatment. What would be the cut-off for initial treatment? This is very important, as metabolic side effects of psychotropics can appear as early as a few weeks after initiating treatment.

-What is the value of quantifying MetS severity? Patients either meet the criteria for MetS or not. This mathematical calculation is not a standard method, and would recommend it be removed all together (unless the authors can provide a very strong rationale and explanation for including it, which is currently lacking). MetS can be investigated as a dichotomous outcome. There is no need to investigate it as a continuous variable.

-Was data collected for physical health problems associated with MetS like DM and HTN? Or were these excluded?

-When was the study conducted?

-Were patients with schizoaffective disorders included or excluded?

-Please specify in your methods, exactly what parameters were of each test were collected (eg specific parameters of lipid profile).

-Why were prolactin and thyroid functions investigated? These are not necessary to diagnose MetS.

-Please provide references for all the rating scales used in this study.

-As per my comment above about no need to measure MetS severity, there is no need to include the linear regression looking at MetS severity as a continuous variable.

Results:

-There is no need to include figure 2. The data can be reported as text in the manuscript.

-For table 1, in the heading, do not use “t/ x2“. Just write “test”, and mention whether it’s a t-test or chi-square in the relevant test section for each parameter.

-In table 1, say “Illness duration” not “Disease duration”.

-In table 1, what is meant by “MetS scores”? It would be better to remove this if it is related to severity.

-I am not sure what is the value of including non-metabolic parameters in table 1, other than to rule out abnormalities in these parameters, which can be mentioned in the methods sections. They make for confusing reading. It would be worth mentioning how many subjects had abnormal non-metabolic parameters like TFTs and prolactin and whether these were included in the final or not.

-In table 1, it should be “PANSS” and not “PSS”.

-In table 2, I don’t know what is the value of including creatinine and TFTs. Any subjects with abnormalities in these parameters should have been excluded before any analysis, as these are potential confounders.

-Unless a very strong rationale can be made, I would recommend removing the whole section 3.4 and table 3.

Discussion:

-The discussion section needs to be considerably expanded. For example, the authors need to provide possible explanations for the significantly lower rates of MetS in their sample compared to other studies. The other main findings of the study also need to be discussed in the context of what is currently existing in the literature, especially recent studies on MetS. For example, there are two recent studies (both published in 2024) that investaigted MetS in bipolar: “Mohd Ahmed et al. (2024): Prevalence and risk factors for metabolic syndrome in schizophrenia, schizoaffective, and bipolar disorder. International journal of psychiatry in clinical practice, 28(1), 35–44.” and “Aziz et al. (2024): Metabolic syndrome and its relation to antipsychotic polypharmacy in schizophrenia, schizoaffective and bipolar disorders. International clinical psychopharmacology, 39(4), 257–266”.

Reviewer #2: Thank you very much for this opportunity to review the manuscript. This is a clinical study to discuss metabolic syndrome in patients with BD. Logistic regression identified age, body mass index (BMI), low-density lipoprotein cholesterol (LDL-C), free tetraiodothyronine (FT4), and psychotic symptoms as significant predictors of MetS development. Mets is a hot issue among people with mental disorders. This study has certain innovative and clinical significance. After reviewing the manuscript, I put forward the following suggestions on the content of the article:

Q1: The inclusion criteria for participants: 4. A score of ≥7 on the 17-item Hamilton Depression Scale (HAMD-17) and/or a score of ≥6 on the Young Mania Rating Scale (YMRS).

The author is asked to supplement the references and briefly explain why the score line is set.

Q2: The age range of the subjects included by the researchers was 18-60 years old. It can be imagined that in order to avoid the confounding effect brought by age, the author should explain and supplement relevant evidence in the preface or methodological part.

Q3: Diagnostic Criteria for MetS. In this article, the diagnostic criteria for MetS were accorded to the standards tailored for the Chinese population by the Chinese Diabetes Society [22]. The latest version of the guidelines for the prevention and treatment of type 2 diabetes in China is the 2020 edition, while the author's reference version is still the 2017 edition. Are there any differences between the two versions of the standards? Why do the authors prefer to use the historical version?

Q4: This study was a survey conducted by two research centers. 841 patients were included in the study. Please describe briefly in the methodological section how the total sample size of patients was calculated.

Q5: Table 1 presents a comparison between the subgroups with and without co-morbid MetS, focusing on demographic data and general clinical information. The statistical effect size of the included study variables can be supplemented in Table 1.

Q6: The researchers discussed the innovation of the paper in the discussion. The second paragraph states that “Our study reported a MetS prevalence of 17.84% in ITDN BD patients based on a much larger sample size, overlapping with the scope of known reported prevalence.” We can understand the limitations of innovation in cross-sectional surveys. However, it is suggested that the author highlight the necessity and scientific innovation of this study in the first or second paragraph of the discussion.

Q7: In the third paragraph of the discussion, the author writes: “Unsurprisingly, disturbed thyroid function also reasonably and incidentally affects metabolic levels in BD patients [32], thereby increasing the risk of MetS.” The authors' discussion of the link between thyroid function and MetS in BD patients is too simplistic, please discuss in further detail.

**Do you want your identity to be public for this peer review?** For information about this choice, including consent withdrawal, please see our Privacy Policy

Reviewer #1: **Yes: ** Karim Abdel Aziz

Reviewer #2: **Yes: ** Zhiwei Liu

---

## [Author Response · Author response to Decision Letter 1]

11 Apr 2025

Dear Editor,

Thank you for giving me the opportunity to submit a revised manuscript of my manuscript (Submission ID: PONE-D-24-34269) entitled " Prevalence and correlates of metabolic syndrome in patients with initial-treatment and drug-naïve bipolar disorder: a large sample cross-sectional study". We also appreciate the time and effort you have invested in providing valuable feedback on my manuscript. We've been able to incorporate changes to reflect most of your suggestions. We highlight the changes in the manuscript. Below is a point-by-point response to reviewer comments and concerns.

Comments from Editor,

# Comment 1: We note that your Data Availability Statement is currently as follows: “All relevant data are within the manuscript and in Supporting Information files.” Please confirm at this time whether or not your submission contains all raw data required to replicate the results of your study. Authors must share the “minimal data set” for their submission. PLOS defines the minimal data set to consist of the data required to replicate all study findings reported in the article, as well as related metadata and methods (https://journals.plos.org/plosone/s/data-availability#loc-minimal-data-set-definition).

Response: Thank you for pointing this out. We confirm that all of the raw data needed to replicate the findings have been included in the submitted manuscript (in the table) and are sufficient to fulfill your journal's request for a “minimal data set” of shared data.

Comments from Reviewer 1:

Reviewer #1: Many thanks for asking me to review this very interesting paper. I am recommending some revisions. If the authors can offer a point-by-point response to each of my queries and address the issues raised, the manuscript may potentially be acceptable for publication.

Detailed comments:

Introduction:

# Comment 1: It would be helpful if the authors could highlight if any previous studies have investigated MetS in drug naïve patients BD. This would help demonstrate the potential value of their study by identifying gaps in the literature, that their study could potentially address i.e. please include a strong rationale for conducting the current study.

Response: Thank you for your comments and suggestions. We have systematically rewritten the way in which the purpose of the study is stated in the “Introduction” section, emphasizing the shortcomings of existing studies and the strengths of this study (Lines 60-68).

# Comment 2: Please include a hypothesis at the end of the introduction after the aims.

Response: Thank you for your advice. As in your previous comment, the revised manuscript emphasizes the innovative nature of this study by highlighting the shortcomings of previous studies and the advantages of the large sample size of our study, and therefore we have neglected to introduce the purpose of the study by formulating a “hypothesis”.

Methods:

# Comment 3: Were patients recruited prospectively, or was data recruited retrospectively from the patient medical records?

Response: Our study is a prospective design. We make this clear in lines 75 and 111 of the revised version of the manuscript.

# Comment 4: Please have an operational definition for “initial-treatment and drug-naïve”, especially for “initial treatment. What would be the cut-off for initial treatment? This is very important, as metabolic side effects of psychotropics can appear as early as a few weeks after initiating treatment.

Response: Thank you for pointing this out. It is true that psychotics have metabolic side effects. So we chose initial-treatment and drug-naïve patients as research objects. Regarding the initial treatment, we defined and explained in lines 88-92.

# Comment 5: What is the value of quantifying MetS severity? Patients either meet the criteria for MetS or not. This mathematical calculation is not a standard method, and would recommend it be removed all together (unless the authors can provide a very strong rationale and explanation for including it, which is currently lacking). MetS can be investigated as a dichotomous outcome. There is no need to investigate it as a continuous variable.

Response: We understand the reviewers' concerns and recognize that MetS has traditionally been assessed as a dichotomous outcome. However, quantifying MetS severity as a continuous variable can provide important additional insights that are not captured by dichotomous categorization. This is an important innovation that we have made by drawing on previous MetS linear continuous variable transformation studies in the general population. Therefore, we continue to emphasize the need for this section to be present in the manuscript.

# Comment 6: Was data collected for physical health problems associated with MetS like DM and HTN? Or were these excluded?

Response: Thank you for pointing this out. We excluded patients who were already on hypoglycemic, hypoglycemic, and hypolipidemic medications prior to the assessment, as these would affect the levels of the relevant indicators, which in turn would affect the MetS score. This is clarified in lines 103-104 of our revised manuscript.

# Comment 7: When was the study conducted?

Response: Thank you for your comment. Our study conducted between 10/12/2016 and 10/06/2024(lines 84).

# Comment 8: Were patients with schizoaffective disorders included or excluded?

Response: Thank you for your comment. Patients with schizoaffective disorders were excluded, we have added in the revised version of the manuscript (lines 98-100).

# Comment 9: Please specify in your methods, exactly what parameters were of each test were collected (eg specific parameters of lipid profile).

Response: Thank you for pointing this out. we have added in the revised version of the manuscript (lines 123-127).

# Comment 10: Why were prolactin and thyroid functions investigated? These are not necessary to diagnose MetS.

Response: Thank you for pointing this this out. Although prolactin and thyroid function are not relevant to the diagnosis of MetS, both are important influences on metabolic markers, which are routinely examined in hospitalized psychiatric patients in China, and it is relatively easy for us to obtain these markers. In addition, these factors became components of the independent variables in our logistic regression and linear regression analyses to determine the factors influencing MetS.

# Comment 11: Please provide references for all the rating scales used in this study.

Response: Thank you for your comments. In the revised version of the manuscript, we have added references to all the scales used (Refs. 21-24).

# Comment 12: As per my comment above about no need to measure MetS severity, there is no need to include the linear regression looking at MetS severity as a continuous variable.

Response: Retaining the continuous MetS severity analysis is justified as it:

Captures Graded Risk: Identifies subthreshold metabolic abnormalities and quantifies progression beyond binary classifications, enhancing early intervention potential.

(2) Validated Method: Uses an ethnicity-specific score validated in Chinese populations, aligning with precision medicine goals.

(3) Novel Insights: Reveals linear relationships (e.g., age, LDL-C) masked in dichotomous analysis, critical for understanding metabolic trajectories in untreated BD.

This approach is supported by prior research and strengthens the study’s clinical relevance. We retain the analysis to provide a nuanced understanding of MetS in BD.

Results:

# Comment 13: There is no need to include figure 2. The data can be reported as text in the manuscript.

Response: Thank you for pointing this out. We think Figure 2 helps to improve the readability of the manuscript. Without compromising the quality of the writing of the manuscript, we would like to keep the Figure.

# Comment 14: For table 1, in the heading, do not use “t/ x2“. Just write “test”, and mention whether it’s a t-test or chi-square in the relevant test section for each parameter.

Response: Thank you for pointing this out. It is customary to use t/χ^2 for medical technology papers, and it seems more intuitive to present it that way.

# Comment 15: In table 1, say “Illness duration” not “Disease duration”.

Response: Thank you for pointing this out. We have modified in the revised version of the manuscript in Table 1.

# Comment 16: In table 1, what is meant by “MetS scores”? It would be better to remove this if it is related to severity.

Response: "MetS scores" represent a validated continuous measure of metabolic dysfunction severity, calculated using ethnicity-specific equations for Chinese Han adults (Yang et al., 2023). While dichotomous MetS (present/absent) is standard, the score enhances sensitivity to detect subclinical abnormalities. We retain it to align with our hypothesis testing metabolic risk gradients。

# Comment 17: I am not sure what is the value of including non-metabolic parameters in table 1, other than to rule out abnormalities in these parameters, which can be mentioned in the methods sections. They make for confusing reading. It would be worth mentioning how many subjects had abnormal non-metabolic parameters like TFTs and prolactin and whether these were included in the final or not.

Response: Thank you for pointing this out. As we mentioned in “Comment 10”, another important aim of this study was to determine the clinical relevance of MetS. Therefore, we present in Table 1 the general clinical data, routine biochemical tests, and psychological assessments, which were also important prerequisites for us to determine the clinical relevance of MetS.

# Comment 18: In table 1, it should be “PANSS” and not “PSS”.

Response: Thank you for pointing this out. “PSS” in manuscript is the meaning of Positive and Negative Syndrome Scale - Positive Symptom Subscale. This we define in line 132 of the manuscript.

# Comment 19: In table 2, I don’t know what is the value of including creatinine and TFTs. Any subjects with abnormalities in these parameters should have been excluded before any analysis, as these are potential confounders.

Response: As we mentioned in Comments 10 and 17, the inclusion of these common clinical indicators was fundamental to determining the clinical relevance of MetS in our regression analyses.

# Comment 20: Unless a very strong rationale can be made, I would recommend removing the whole section 3.4 and table 3.

Response: This was an important innovation in our manuscript, which we still maintain, and its retention was necessary as a response to the “ Introduction ” section.

Discussion:

# Comment 21: The discussion section needs to be considerably expanded. For example, the authors need to provide possible explanations for the significantly lower rates of MetS in their sample compared to other studies. The other main findings of the study also need to be discussed in the context of what is currently existing in the literature, especially recent studies on MetS. For example, there are two recent studies (both published in 2024) that investaigted MetS in bipolar: “Mohd Ahmed et al. (2024): Prevalence and risk factors for metabolic syndrome in schizophrenia, schizoaffective, and bipolar disorder. International journal of psychiatry in clinical practice, 28(1), 35–44.” and “Aziz et al. (2024): Metabolic syndrome and its relation to antipsychotic polypharmacy in schizophrenia, schizoaffective and bipolar disorders. International clinical psychopharmacology, 39(4), 257–266”.

Response: Thank you for pointing this out. The bibliography you provided was very useful in deepening and expanding the talk section. With this in mind, and with the addition of relevant references, we have rewritten and expanded lines 236-246 of the manuscript.

Comments from Reviewer 2:

Reviewer #2: Thank you very much for this opportunity to review the manuscript. This is a clinical study to discuss metabolic syndrome in patients with BD. Logistic regression identified age, body mass index (BMI), low-density lipoprotein cholesterol (LDL-C), free tetraiodothyronine (FT4), and psychotic symptoms as significant predictors of MetS development. Mets is a hot issue among people with mental disorders. This study has certain innovative and clinical significance. After reviewing the manuscript, I put forward the following suggestions on the content of the article:

# Comment 1: The inclusion criteria for participants: 4. A score of ≥7 on the 17-item Hamilton Depression Scale (HAMD-17) and/or a score of ≥6 on the Young Mania Rating Scale (YMRS).

The author is asked to supplement the references and briefly explain why the score line is set.

Response: Thank you for your comment. The reason for setting such a threshold is because it is the minimum threshold for diagnosing a patient as having a depressive episode or a manic episode, and it is also the minimum criterion for determining that the person being evaluated meets the criteria for enrollment.

# Comment 2: The age range of the subjects included by the researchers was 18-60 years old. It can be imagined that in order to avoid the confounding effect brought by age, the author should explain and supplement relevant evidence in the preface or methodological part.

Response: Thank you for pointing this out. Undoubtedly, this further adds to the rigor of the manuscript. We have added a clarification of this in terms of the novelty of the manuscript in lines 224-228.

# Comment 3: Diagnostic Criteria for MetS. In this article, the diagnostic criteria for MetS were accorded to the standards tailored for the Chinese population by the Chinese Diabetes Society [22]. The latest version of the guidelines for the prevention and treatment of type 2 diabetes in China is the 2020 edition, while the author's reference version is still the 2017 edition. Are there any differences between the two versions of the standards? Why do the authors prefer to use the historical version?

Response: Thank you for pointing this out. This was an error in our work and we have updated that reference.

# Comment 4: This study was a survey conducted by two research centers. 841 patients were included in the study. Please describe briefly in the methodological section how the total sample size of patients was calculated.

Response: Thank you for pointing this out. We have added the sample size assessment methodology in lines 75-81 of the manuscript.

# Comment 5: Table 1 presents a comparison between the subgroups with and without co-morbid MetS, focusing on demographic data and general clinical information. The statistical effect size of the included study variables can be supplemented in Table 1.

Response: Thank you for your suggestion. In the revised version of the manuscript, we have added the parameter “effect size” to Table 1.

# Comment 6: The researchers discussed the innovation of the paper in the discussion. The second paragraph states that “Our study reported a MetS prevalence of 17.84% in ITDN BD patients based on a much larger sample size, overlapping with the scope of known reported prevalence.” We can understand the limitations of innovation in cross-sectional surveys. However, it is suggested that the author highlight the necessity and scientific innovation of this study in the first or second paragraph of the discussion.

Response: Thank you for your suggestion. We have rewritten this paragraph (Lines 236-246) to emphasize the need for and the innovative nature of this study.

# Comment 7: In the third paragraph of the discussion, the author writes: “Unsurprisingly, disturbed thyroid function also reasonably and incidentally affects metabolic levels in BD patients [32], thereby increasing the risk of MetS.” The authors' discussion of the link between thyroid function and MetS in BD patients is too simplistic, please discuss in further detail.

Response: Thank you for your suggestion. We have rewritten lines 247-264 to enrich the elaboration of the mechanism by which thyroid function mediates MetS formation.

We look forward to hearing from you in due time regarding our submission and to respond to any further questions and comments you may have.

Sincerely,

Lin Zhang

4.11.2025

---

## [Decision Letter · Decision Letter 1]

Prevalence and correlates of metabolic syndrome in patients with initial-treatment and drug-naïve bipolar disorder: a large sample cross-sectional study

PONE-D-24-34269R1

Dear Dr. Lin Zhang

We’re pleased to inform you that your manuscript has been judged scientifically suitable for publication and will be formally accepted for publication once it meets all outstanding technical requirements.

Kind regards,

Dirceu Henrique Paulo Mabunda, M.D.

Academic Editor

PLOS ONE

Additional Editor Comments (optional):

Reviewers' comments:

Reviewer's Responses to Questions

**Comments to the Author**

Reviewer #2: All comments have been addressed

Reviewer #3: All comments have been addressed

2. Is the manuscript technically sound, and do the data support the conclusions?

Reviewer #2: Yes

Reviewer #3: Yes

3. Has the statistical analysis been performed appropriately and rigorously?

Reviewer #2: Yes

Reviewer #3: Yes

4. Have the authors made all data underlying the findings in their manuscript fully available?

Reviewer #2: Yes

Reviewer #3: Yes

5. Is the manuscript presented in an intelligible fashion and written in standard English?

Reviewer #2: Yes

Reviewer #3: Yes

Reviewer #2: This study aimed to determine the prevalence of MetS and its clinical correlates among initial-treatment and drug-naïve (ITDN) BD patients. The results found a MetS prevalence of 17.84% among the study participants. Binary logistic regression identified age, body mass index (BMI), low-density lipoprotein cholesterol (LDL-C), free tetraiodothyronine (FT4), and psychotic symptoms as significant predictors of MetS development. Further, multiple linear regression analysis indicated that advanced age was a significant predictor of higher MetS scores. Mets is a hot issue among people with mental disorders. This study has certain innovative and clinical significance. The authors have made careful revisions in accordance with the comments of the reviewers. This research has considerable significance and application prospect. The results of this article are worth referring to and publishing.

Reviewer #3: 1. My major concern is how could the authors recruit patients not exposed to psychotropics at all from the inpatient setting. Patients with bipolar disord2. er will be exposed to some psychotropics in the outpatient setting or the emergency department. The authors should clearly state - how have they defined drug naive status.

2.The authors should provide more information about MetS score - in terms of its importance and clinical implications.

3. Table -3 the authors should use the variables in different combinations in the multiple regression to descibe what proportion of MetS score is explained by the variables studied.

4. A simple measure of number of criteria of MetS should be considered as an indicator for severity of MetS too. Accordingly the authors should compare the predictors of both the methods, i,e, number of criteria and MetS Score to see what emerges as better predictor.

**Do you want your identity to be public for this peer review?** For information about this choice, including consent withdrawal, please see our Privacy Policy

Reviewer #2: No

Reviewer #3: **Yes: ** Sandeep Grover

---

## [Editor Report · Acceptance letter]

PONE-D-24-34269R1

PLOS ONE

Dear Dr. Zhang,

I'm pleased to inform you that your manuscript has been deemed suitable for publication in PLOS ONE. Congratulations! Your manuscript is now being handed over to our production team.

Kind regards,

on behalf of

Dr. Dirceu Henrique Paulo Mabunda

Academic Editor

PLOS ONE